# Zika virus infection drives epigenetic modulation of immunity by the histone acetyltransferase CBP of *Aedes aegypti*

Anderson de Mendonça Amarante[1,2], Isabel Caetano de Abreu da Silva[1,2], Vitor Coutinho Carneiro[3], Amanda Roberta Revoredo Vicentino[1], Marcia de Amorim Pinto[1], Luiza Mendonça Higa[4], Kanhu Charan Moharana[5], Octavio A. C. Talyuli[1,2], Thiago Motta Venancio[2,5], Pedro Lagerblad de Oliveira[1,2], Marcelo Rosado Fantappié[1,2] *

**1** Instituto de Bioquímica Médica Leopoldo de Meis, Programa de Biologia Molecular e Biotecnologia, Centro de Ciências da Saúde, Universidade Federal do Rio de Janeiro, Rio de Janeiro, Brasil, **2** Instituto Nacional de Entomologia Molecular, Universidade Federal do Rio de Janeiro, Rio de Janeiro, Brasil, **3** Division of Epigenetics, German Cancer Research Center, Im Neuenheimer Feld, Heidelberg, Germany, **4** Departamento de Genética, Instituto de Biologia, Universidade Federal do Rio de Janeiro, Rio de Janeiro, Brasil, **5** Laboratório de Química e Função de Proteínas e Peptídeos, Centro de Biociências e Biotecnologia, Universidade Estadual do Norte Fluminense Darcy Ribeiro, Campos dos Goytacazes, Brasil

* fantappie@bioqmed.ufrj.br

**Data Availability Statement:** All relevant data are within the manuscript and its Supporting Information files.

## Abstract

Epigenetic mechanisms are responsible for a wide range of biological phenomena in insects, controlling embryonic development, growth, aging and nutrition. Despite this, the role of epigenetics in shaping insect-pathogen interactions has received little attention. Gene expression in eukaryotes is regulated by histone acetylation/deacetylation, an epigenetic process mediated by histone acetyltransferases (HATs) and histone deacetylases (HDACs). In this study, we explored the role of the *Aedes aegypti* histone acetyltransferase CBP (AaCBP) after infection with Zika virus (ZIKV), focusing on the two main immune tissues, the midgut and fat body. We showed that the expression and activity of AaCBP could be positively modulated by blood meal and ZIKV infection. Nevertheless, Zika-infected mosquitoes that were silenced for AaCBP revealed a significant reduction in the acetylation of H3K27 (CBP target marker), followed by downmodulation of the expression of immune genes, higher titers of ZIKV and lower survival rates. Importantly, in Zika-infected mosquitoes that were treated with sodium butyrate, a histone deacetylase inhibitor, their capacity to fight virus infection was rescued. Our data point to a direct correlation among histone hyperacetylation by AaCBP, upregulation of antimicrobial peptide genes and increased survival of Zika-infected-*A. aegypti*.

## Author summary

Pathogens have coevolved with mosquitoes to optimize transmission to hosts. As natural vectors, mosquitoes are permissive to and allow systemic and persistent arbovirus infection, which intriguingly does not result in dramatic pathological sequelae that affect their

**Funding:** This work was supported by grants from Conselho Nacional de Desenvolvimento Científico e Tecnológico (470099/20143) and Fundação Carlos Chagas Filho de Amparo a Pesquisa do Estado do Rio de Janeiro (E-26/202990/2015) to MRF. PLO was supported by Instituto Nacional de Ciência e Tecnologia em Entomologia Molecular (573959/2008-0). The funders had no role in study design, data collection and analysis, decision to publish, or preparation of the manuscript.

**Competing interests:** The authors have declared that no competing interests exist.

lifespan. In this regard, mosquitoes have evolved mechanisms to tolerate persistent infection and develop efficient antiviral strategies to restrict viral replication to nonpathogenic levels. There is a great deal of evidence supporting the implication of epigenetics in the modulation of the biological interaction between hosts and pathogens. This study reveals that Zika virus infection positively modulates the expression and activity of *A. aegypti* histone acetyltransferase CBP (AaCBP). This study shows that AaCBP plays a role in the activation of immune-responsive genes to limit Zika virus replication. This first description that Zika virus infection has epigenomic consequences in the regulation of *A. aegypti* immunity opens a new avenue for research on mosquito factors that can drive vector competence.

## Introduction

Mosquitoes are primary vectors of a variety of human pathogens throughout the world. *Aedes aegypti* mosquitoes can develop long-lasting viral infections and carry high viral loads, which make them efficient vectors for the transmission of arboviruses such as Zika virus (ZIKV) [1].

Host-pathogen interactions are among the most plastic and dynamic systems in nature. In this regard, epigenetic modifications can offer an accessory source of fast-acting, reversible and readily available phenotypic variation that can be directly shaped by both host and pathogen selection pressures [2,3]. One of the hallmarks in the study of host gene regulation is to elucidate how specific sets of genes are selected for expression in response to pathogen infection.

Understanding the interactions between the mosquito immune system and viruses is critical for the development of effective control strategies against these diseases. Mosquitoes have conserved immune pathways that limit infections by viral pathogens [4,5]. Mosquito antiviral defense is regulated by RNA interference (RNAi), Janus kinase/signal transducer (JAK-STAT), Toll, the immune deficiency (IMD) and MAPK immune pathways [4–6].

The Toll pathway has been shown to play the most important role in controlling ZIKV infections [4]. In this context, gene expression analysis of ZIKV-infected mosquitoes has indicated that Toll pathway-related genes are highly upregulated in Zika infection when compared to other immune pathways [4,5].

Eukaryotic gene expression is controlled by the functions of *cis*-DNA elements, enhancers and promoters, which are bound by transcription factors, in combination with the organization of the chromatin [7]. Although it is still poorly understood how chromatin-associated processes participate in the regulation of gene transcription in the context of pathogen-vector interactions, it has been previously shown that *Plasmodium falciparum* infection induces significant chromatin changes in the *Anopheles gambiae* mosquitoes [3]. This study identified infection-responsive genes showing differential enrichment in various histone modifications at the promoter sites [3].

The transcriptional coactivator CREB-binding protein (CBP), and its paralog p300, play a central role in coordinating and integrating multiple signal-dependent events with the transcription apparatus, allowing the appropriate level of gene activity to occur in response to diverse stimuli [8]. CBP proteins do not specifically interact with the promoter elements of target genes, but they are recruited to promoters by interaction with DNA-bound transcription factors, which directly interact with the RNA polymerase II complex [8,9]. A key property of CBP is the presence of histone acetyltransferase (HAT) activity, which endows the enzyme with the capacity to influence chromatin activity by the acetylation of histones [8].

Consistent with its function as a transcriptional coactivator, CBP plays crucial roles in embryogenesis [10], development [11], differentiation [11,12], oncogenesis [11,13] and immunity [9]. Although most of these studies have been conducted in mammalian systems, nonvector insect models have also made important contributions to the biological functions of CBP [14–17].

The present work describes the functional characterization of a histone acetyltransferase in an insect vector and provides indications that Zika virus infection exerts epigenomic consequences in regulating *A. aegypti* immunity.

## Materials and methods

### Ethics statement

All animal care and experimental protocols were conducted in accordance with the guidelines of the Committee for Evaluation of Animal Use for Research–CEUA of the Federal University of Rio de Janeiro (UFRJ). The protocols were approved by CEUA-UFRJ under the registration number IBQM149/19. Technicians in the animal facility at the Instituto de Bioquímica Médica Leopoldo de Meis (UFRJ) carried out all protocols related to rabbit husbandry under strict guidelines to ensure careful and consistent animal handling.

### Mosquito rearing and cell culture

*Aedes aegypti* (*Liverpool black-eyed* strain) were raised in a mosquito rearing facility at the Federal University of Rio de Janeiro, Brazil, under a 12-h light/dark cycle at 28°C and 70–80% relative humidity. Larvae were fed dog chow, and adults were maintained in a cage and given a solution of 10% sucrose *ad libitum*. Females 7–10 days posteclosion were used in the experiments. When mentioned, mosquitoes were artificially fed heparinized rabbit blood. For virus infection, mosquitoes were fed using water-jacketed artificial feeders maintained at 37°C sealed with parafilm membranes. Alternatively, mosquitoes were infected by intrathoracic injections of 69 nL containing 60 Plaque Forming Units (PFUs) of ZIKV.

Female midguts or fat bodies were dissected 24, 48, 72 or 96 h after feeding for RNA sample preparation.

The *A. aegypti* embryonic cell line Aag2 was maintained in Schneider medium (Merk) supplemented with 5% FBS (LGC, Brazil) and 1% penicillin/streptomycin/amphotericin B. Aag2 cells were incubated at 28°C.

### ZIKV infection and virus titration

ZIKV strain Pernambuco (ZIKV strain ZIKV/H.sapiens/Brazil/PE243/201) [18] was propagated in the *A. albobictus* C6/36 cell line, and titers were determined by plaque assay on Vero cells. ZIKV was propagated in C6/36 cells for 6–8 days; virus was then harvested, filtered through 0.22 and 0.1μm membranes, respectively and stored at -80°C. Mosquitoes were intrathoracically infected by microinjections [19] of 69 nL of virus, containing 60 PFUs.

Midguts and fat bodies were dissected, individually collected, and stored at– 80°C until use for plaque assays. Virus titration was performed as described previously [20]. Plates were incubated for 4–5 days, fixed and stained with a methanol/acetone and 1% crystal violet mixture, and washed, after which the plaque forming units (PFUs) were counted.

### AaCBP gene knockdown by RNAi

Double-stranded RNA (dsRNA) was synthesized from templates amplified from cDNA of adult female mosquitoes using specific primers containing a T7 tail (S1 Table). The *in vitro*

dsRNA transcription reaction was performed following the manufacturer's instructions (Ambion MEGAscript RNAi). Two different dsRNA products (dsAaCBP1 and dsAaCBP2) were PCR amplified based on the coding sequence of *A. aegypti* CBP (GenBank accession number XP_011493407.2), using the oligonucleotides listed in S1 Table. The irrelevant control gene luciferase (dsLuc) was amplified from the luciferase T7 control plasmid (Promega). Female mosquitoes were injected intrathoracically [21] with dsRNA (0.4 μg) with a microinjector (NanoJect II Autonanoliter injector, Drummond Scientific, USA). Injected mosquitoes were maintained at 28°C, and 70–80% humidity, with 10% sucrose provided *ad libitum*.

### Nucleic acids isolation and quantitative real-time PCR

Total RNA was isolated from whole bodies, midguts or fat bodies of adult females, using the RiboPure kit (Ambion) followed by DNase treatment (Ambion) and cDNA synthesis (SuperScript III First-Strand Synthesis System, Invitrogen), following the manufacturer's instructions. Quantitative reverse transcription gene amplifications (qRT-PCR) were performed with StepOnePlus Real-Time PCR System (Applied Biosystems) using the Power SYBR Green PCR Master Mix (Applied Biosystems). The comparative *Ct* method [22] was used to compare mRNA abundance. In all qRT-PCR analyses, the *A. aegypti* ribosomal protein 49 gene (*Rp*49) was used as an endogenous control [23]. All oligonucleotide sequences used in qRT-PCR assays are listed in the S1 Table.

The content of the mosquito intestinal microbiota was analyzed by genomic DNA qPCR. Genomic DNA was isolated using the DNeasy Blood and Tissue Kit (Qiagen), following the manufacturer's instructions. Amplifications were carried out using specific primers of the ribosomal genes 16S and 18S (S1 Table), for bacteria or fungi, respectively. The *A. aegypti* ribosomal protein 49 gene (*Rp*49) was used as an endogenous control. Quantifications were carried out using the comparative *Ct* method. Fifty nanograms of genomic DNA from a pool of 15 midguts from female mosquitoes was used as a template. For midgut dissection, mosquitoes were sterilized in 20% hypochlorite solution, followed by 70% ethanol for 30 seconds in each solution. Sterilized midguts were washed in sterile 1X PBS solution.

### Western blotting

Protein extracts were prepared as previously described [24,25]. Briefly, *A. aegypti* total protein extracts were carried out by homogenizing adult female mosquitoes or Aag2 cells in TBS containing a protease inhibitor cocktail (Sigma). Proteins were recovered from the supernatant by centrifugation at 14.000xg, for 15 min. at 4°C. Protein concentration was determined by the Bradford Protein Assay (Bio-Rad). Western blots were carried out using secondary antibody (Immunopure goat anti-mouse, #31430). The primary monoclonal antibodies (ChIP grade) used were anti-H3 pan acetylated (Sigma-Aldrich #06–599), anti-H3K9ac (Cell Signaling Technology #9649) and Anti-H3K27ac (Cell Signaling Technology, #8173), according to the manufacture's instructions. For all antibodies, a 1:1000 dilution was used. For normalization of the signals across the samples, an anti-histone H3 antibody (Cell Signaling Technology, #14269) was used.

### Statistical analysis

Survival curves were generated with the Kaplam-meier analysis using the log-rank test. All other analyses were performed with the GraphPad Prism statistical software package (Prism version 6.0, GraphPad Software, Inc., La Jolla, CA). *Asterisks* indicate significant differences (*, $p < 0.05$; **, $p < 0.01$; ***, $p < 0.001$; ns, not significant).

## Genome-wide identification of lysine acetyltransferases in *A. aegypti*

We obtained the latest *A. aegypti* proteome and functional annotations (AaegL5.0) from NCBI RefSeq (https://ftp.ncbi.nlm.nih.gov/genomes/all/GCF/002/204/515/GCF_002204515.2_AaegL5.0). The amino acid sequences of the 23 *D. melanogaster* lysine acetyltransferases (DmKATs) reported by Feller *et al* 2015 were obtained from FlyBase. The KAT orthologs in *A. aegypti* were identified using BlastP (e-value ≤1e-10, identity ≥25% and query coverage ≥50%) [26]. We further validated the presence of various acetyltransferase domains in these KATs. Conserved domains were predicted in putative KATs using hmmer (e-value ≤ 0.01; http://hmmer.org/) and the PFAM-A database. Following the nomenclature of DmKATs and the conserved domain architectures, AeKATs were classified into 5 major sub-families: HAT1, Tip60, MOF, HBO1, GCN5 and CBP [27]. Conserved domain architectures were rendered with DOG (Domain Graph, version 1.0) [28]. Multiple sequence alignment of AeKATs was performed using using Clustalw [29].

## Results

### *Aedes aegypti* has an ortholog of the CREB-binding protein (CBP) and five additional putative histone acetyltransferases

The *A. aegypti* transcriptional coactivator CBP (AaCBP) contains all five canonical structural and functional domains (Fig 1) of the CBP family [30]. The TAZ, KIX and CREB are the protein interaction domains that mediate interactions with transcription factors. The histone acetyltransferase (HAT) catalytic domain of the CBP enzymes shows high level of conservation among different species (Fig 1), as well as among other HATs from *A. aegypti* (S1A and S1B Fig).

**Fig 1. Overview of AaCBP protein domain conservation.** Schematic representation of the full-length AaCBP protein (XP_011493407.2), depicting the conserved functional domains: the TAZ domain (orange), KIX domain (pink), Bromo domain (blue), HAT domain (purple) and CREB domain (green). The full-length CBP from *Homo sapiens* (NP_001420.2), *Apis melifera* (XP_026294861.1) and *Drosophila melanogaster* (AAB53050.1) are also shown for comparison. The percentages of similarity of the CBP-HAT domains are shown within the purple boxes.

AaCBP also contains a bromo domain, which binds acetylated lysines. The conserved domains are connected by long stretches of unstructured linkers.

Because transcriptional coactivator complexes are intricate structures composed of multiple subunits, and because cooperative assembly of histone acetyltransferases is a rate-limiting step in transcription activation, we aimed to identify other HATs in *A. aegypti*. By searching the latest *A. aegypti* proteome and functional annotations (see Methods), we identified six other putative HATs, all containing the conserved HAT functional domain (S1A and S1B Fig).

## ZIKV infection modulates the expression and activity of AaCBP

Mosquitoes naturally acquire viral infections when they feed on blood. In this context, it is well established that in addition to extensive genome-wide transcriptional modulation in mosquitoes after blood meals [31], viral infections can also modulate host cell gene expression and influence cellular function. Thus, we first evaluated whether a blood meal was able to modulate the expression of AaCBP in the midgut or fat body (S2 Fig). We showed that the expression of AaCBP in both tissues was significantly upregulated by blood meal, reaching its peak of transcription at 24 and 48 h after feeding (S2A and S2B Fig). The stimulated transcription of AaCBP was especially dramatic in the fat body 48 h after a blood meal (S2B Fig). Because we were particularly interested in evaluating the functional role of AaCBP in ZIKV infection, we were forced to change our virus infection approach. As an alternative to feeding infected blood, mosquitoes were infected with ZIKV by intrathoracic injections. Importantly, we showed that ZIKV infection was also able to upregulate the expression of AaCBP in the midgut or fat body, reaching its peak transcription at four days after infection (Fig 2A and 2B). However, we did not see upregulation when we assayed head and thorax (Fig 2C). A similar phenomenon was observed when we used Aag2 cells, where the expression of AaCBP reached their peaks at 6 and 15 h, post infection, respectively (Fig 2D). Importantly, we confirmed that the increase in mRNA expression correlated with the increase in AaCBP acetylation activity (Fig 2E and 2F). Of note, ZIKV infection enhanced the acetylation activity of AaCBP toward lysine 27 of histone H3 (H3K27ac), but not toward lysine 9 of histone H3 (H3K9ac), (Fig 2F). Considering that H3K27 is the main target substrate for CBP enzymes [32], and that H3K9 is the main substrate of Gcn5 HAT [33] (also present in *A. aegypti*; see S1A and S1B Fig), these results point to a specific enhancement of AaCBP by ZIKV infection.

## AaCBP plays a role in the defense and survival of ZIKV-infected mosquitoes

To investigate the role of AaCBP in the lifespan of mosquitoes infected with ZIKV, we knocked down the AaCBP gene two days before infection and followed their survival rates for 20 days, on a daily basis (Fig 3A). It is important to emphasize that it was mandatory to use intrathoracic injections for ZIKV infection in these experiments. This was due to the fact that AaCBP gene silencing was not successfully achieved when a blood meal was utilized (S3 Fig; of note, we showed that the expression of AaCBP is upregulated by blood meal, and thus, this likely counteracted the effects of the silencing). The levels of silencing of AaCBP at Day 2 post dsRNA injections were 50% in the midgut and 80% in the fat body, as judged by its mRNA expression and activity (S4A and S4B Fig). Mock-infected mosquitoes that received dsLuc injections, showed a high rate of survival until Day 15, when survival started to decline, most likely due to normal aging (Fig 3B, black lines. Mock-infected mosquitoes that were silenced for AaCBP revealed a mortality rate of 60% at Day 20 (Fig 3B, blue line), a similar pattern observed for ZIKV-infected mosquitoes that received injections with the dsLuc control (Fig 3B, green line). Importantly, mosquitoes that were silenced for AaCBP followed by ZIKV

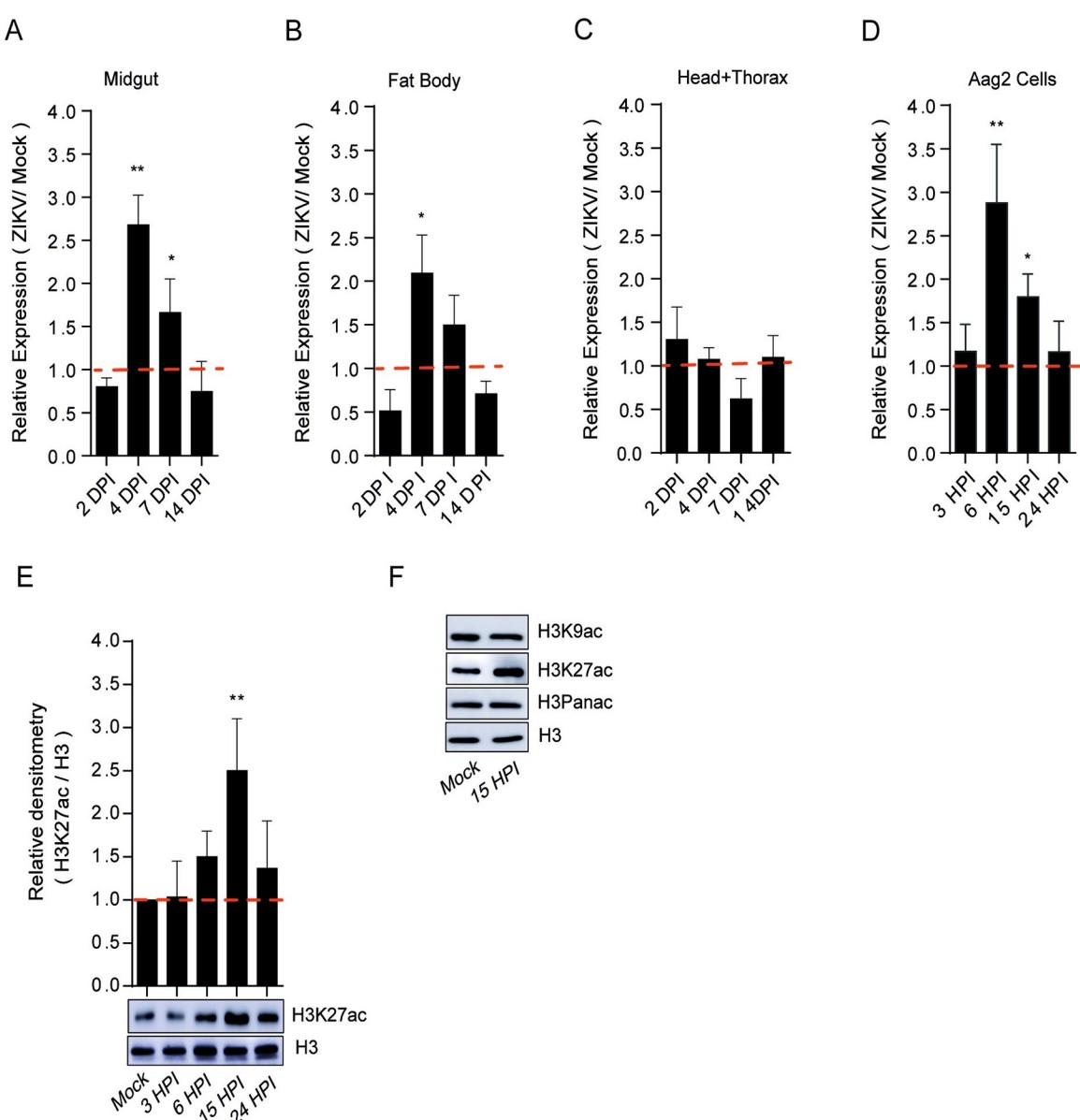

**Fig 2. ZIKV infection modulates the expression and activity of AaCBP.** Aag2 cells as well as mosquitoes were infected with ZIKV at a MOI of 2.0, and 60 PFUs, respectively. Mosquito infections were performed by intrathoracic injections. The expression of AaCBP in mosquitoes (A-C) or Aag2 cells (D) was measured by qRT-PCR on the indicated tissues and days post infection. E and F. Western blot of total protein extract from Aag2 cells infected with ZIKV virus, or mock-infected at different time points. Monoclonal antibodies against H3K9ac, H3K27ac, H3 panacetylated (Panac), or H3 (as a loading control) were used. The intensity of the bands was quantified by densitometry analysis plotted as a graph using ImageJ (NIH Software). Western blotting was performed on 3 independent biological replicates and one representative is shown here. Error bars indicate the standard error of the mean; statistical analyses were performed by Student's *t* test. *, $p < 0.05$; **, $p < 0.01$.

infections showed a much higher mortality rate than all other groups (Fig 3B, orange line). These results indicate that even a partial deletion of AaCBP is enough to disrupt the homeostasis of the mosquito (Fig 3B blue line), and when virus infection occurs, the lack of AaCBP becomes enormously detrimental.

To correlate the lack of AaCBP and mortality, with an increase in viral loads, we determined viral titers over time in the midguts and fat bodies (Fig 3C and 3D, respectively). We

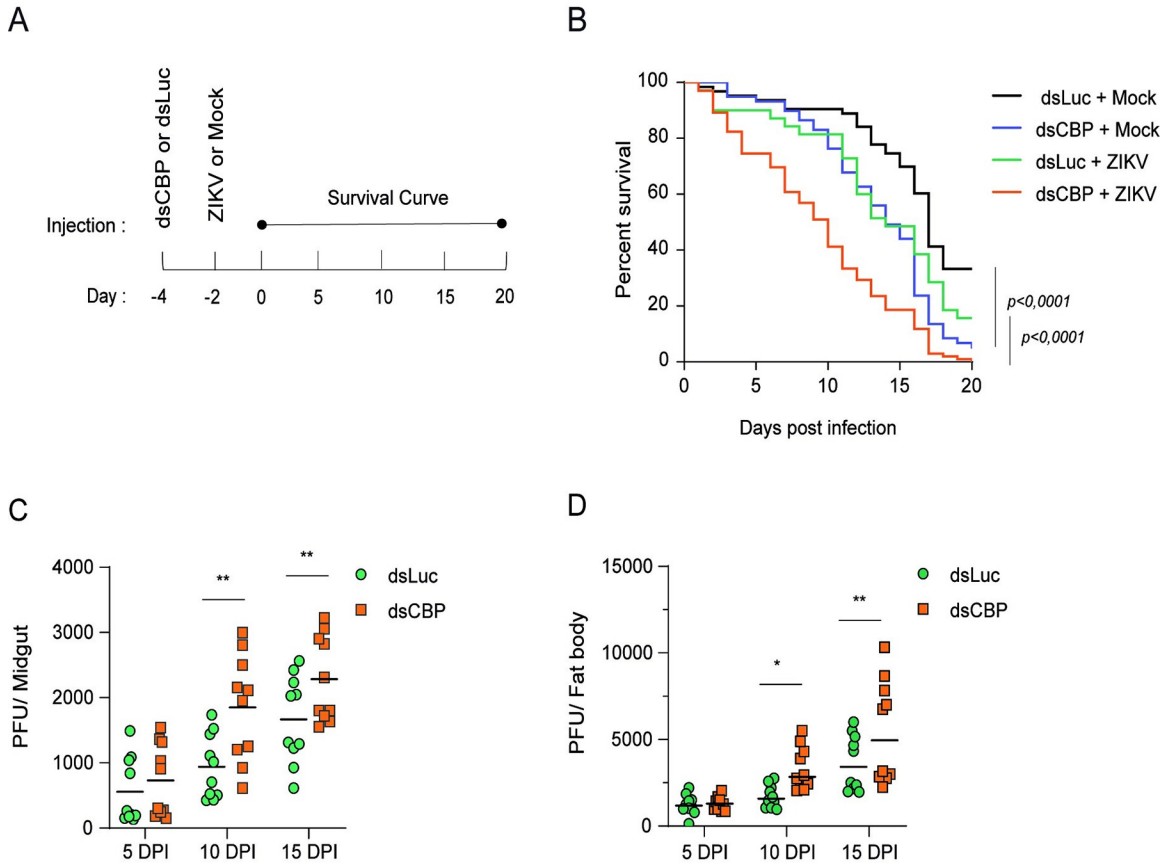

**Fig 3. AaCBP gene knockdown compromises the lifespan of ZIKV-infected mosquitoes.** A. One hundred adult female mosquitoes were injected intrathoracically with double-stranded RNAs for the AaCBP or Luc control gene, two days before Zika infections (by intrathoracic injections). Survival of the mosquitoes was monitored on a daily basis, until Day 20. B. The experiments were conducted in five 500ml cups, each containing 20 mosquitoes. Dead mosquitoes were removed on a daily basis. The survival curves were repeated at least five times. Survival curve of mosquitoes that were not injected, mock-infected mosquitoes that were injected with for dsLuc or dsAaCBP, and mosquitoes that were injected with for dsLuc or dsAaCBP and infected with ZIKV. C and D. Midguts or fat bodies from silenced mosquitoes were assessed for virus infection intensity at different time points. Each dot or square represents the mean plaque-forming units (PFUs) per two midguts or fat bodies (totaling 20 midguts or fat bodies examined) from five independent experiments. Bars indicate the standard error of the mean; statistical analyses were performed by Student's *t* test. *, $p < 0.05$; **, $p < 0.01$.

saw that AaCBP-silenced mosquitoes did not efficiently fight virus infections in these tissues that are expected to mount strong immune responses against viruses [4,5,34]. In this respect, our data reconfirmed that antimicrobial peptides (AMPs) are highly upregulated by ZIKV infections in the midgut (S5 Fig).

## AaCBP regulates the expression of antiviral immune-response genes

Because AaCBP-silenced mosquitoes revealed higher viral loads than control-silenced mosquitoes (Fig 3C and 3D), and knowing the transcriptional coactivator role of CBP enzymes, we questioned whether AaCBP knockdown could have affected the transcription of antiviral immune-response genes. Indeed, qRT-PCR analysis showed that the lack of AaCBP led to downregulation of the immune-response genes *cecropin D, cecropin G, defensin A and defensin C*, in the midgut or fat body (Fig 4A and 4B, respectively). In addition, the *vago 2* gene was also downregulated in the fat body (Fig 4B), but not *defensin A*. Interestingly, *vago 2* has been shown to be upregulated in *A. aegypti* larvae exposed to dengue virus (DENV) [35], as well as

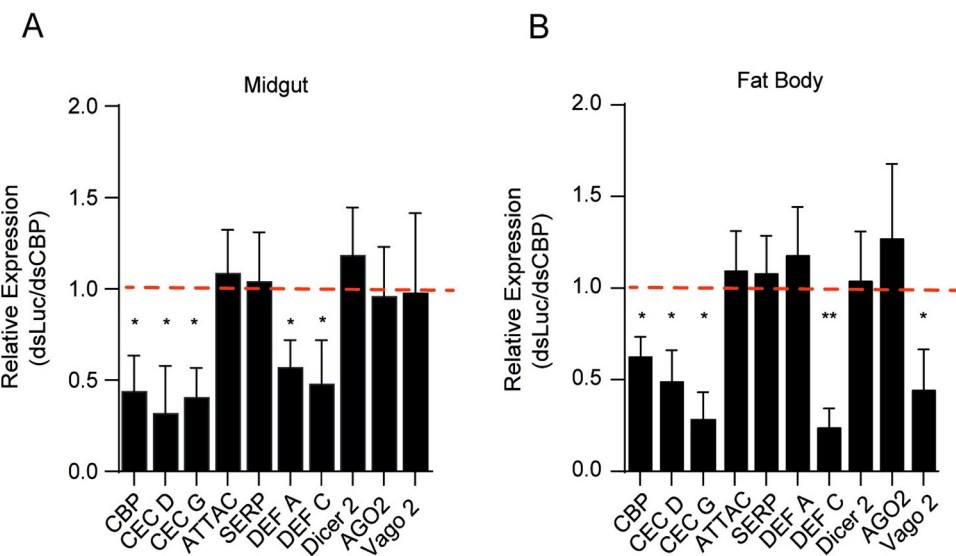

**Fig 4. The expression of antiviral immune-response genes is reduced in ZIKV-infected mosquitoes silenced for AaCBP.** A and B. Fifty adult female mosquitoes were silenced for AaCBP, and after two days, mosquitoes were infected with ZIKV. The expression of *cecropin D* (CEC D), *cecropin G* (CEC G), *attacin* (ATTAC), *serpin* (SERP), *defensin A* (DEF A), *defensin C* (DEF C), *dicer 2*, *argonaute 2* (AGO2) and *vago 2* in the midgut (A) or fat body (B) was measured by qRT-PCR four days post infection. Silencing levels of AaCBP in both tissues are shown (first bars in Panels A and B). qRT-PCR was performed from 3 independent biological replicates. Bars indicate the standard error of the mean; statistical analyses were performed by Student's *t* test. *, $p < 0.05$.

in *A. albopictus* cells infected with DENV [36]. Of note, two members of the RNAi (*si*RNA) pathway, *dicer 2* and *argonaute 2* were not modulated in either tissue of AaCBP-silenced mosquitoes (Fig 4A and 4B). Importantly, downregulation of antiviral immune-response genes persisted until at least day five-post infection, which matched the kinetics of AaCBP-mediated gene silencing (S6A–S6D Fig).

There is always the possibility that the observed dysregulation of immune genes in AaCBP-silenced mosquitoes was accompanied by intestinal dysbiosis. To rule out this possibility, we carried out genomic DNA qPCR to quantify the total bacterial or fungal load in the midguts of AaCBP-silenced or control mosquitoes (S7 Fig). We showed that the midgut microbiota was not affected by AaCBP-silencing (S7 Fig).

## Histone hyperacetylation induces mosquito immune responses and suppression of ZIKV infection

Histone deacetylation by histone deacetylases (HDACs) is involved in chromatin compaction and gene repression [37]. Inhibition of HDACs by sodium butyrate (NaB) leads to histone hyperacetylation and potent gene activation [38]. We treated mosquitoes with NaB and showed a significant increase in H3K27 and H3K9 acetylation (S8A and S8B Fig). An increase in total histone H3 acetylation could not be observed (S8A and S8B Fig). Although we do not have a clear interpretation of this result, we can only speculate at this stage that the levels of the specific acetylated lysines can vary among the different tissues of the whole mosquito, since we are analyzing specific marks (K9 or K27) unlike the other five H3 lysine marks (K4, K14, K18, K23 AND K56).

We next investigated the effect of NaB in ZIKV-infected mosquitoes (Fig 5) and showed that the expression levels of *defensin A*, *defensin C* and *cecropin D* were significantly increased

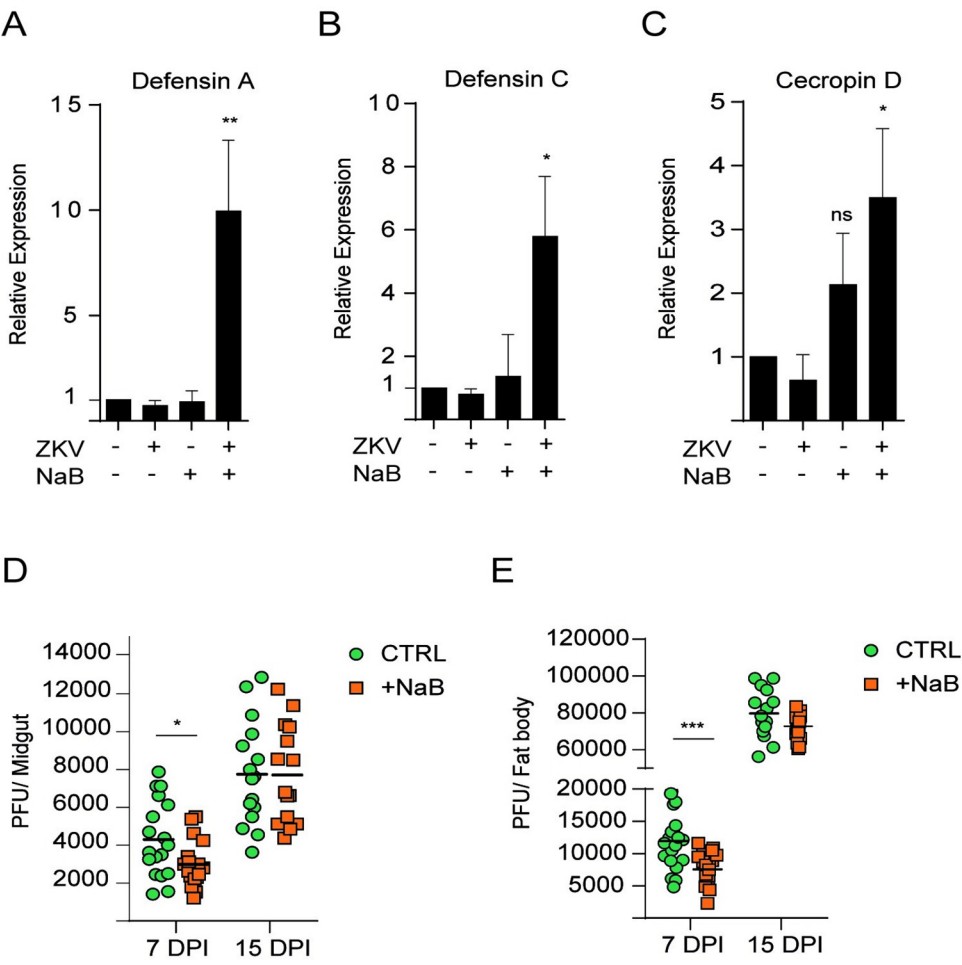

**Fig 5. Histone hyperacetylation in ZIKV-infected mosquitoes induces immune responses and suppresses virus infection.** A-E. Fifty adult female mosquitoes were infected with ZIKV and treated with 0.5 M NaB for three days. A-C. The expression of *defensin A*, *defensin C*, *and cecropin D*, in whole mosquito tissues was measured by qRT-PCR. qRT-PCR was performed on 3 independent biological replicates. D and E. Midguts or fat bodies from ZIKV-infected mosquitoes treated with NaB were assessed for infection intensity at 7 or 15 days post infection. Bars indicate the standard error of the mean; statistical analyses were performed by Student's *t* test. *, $p < 0.05$; **, $p < 0.01$.

(Fig 5A–5C). Importantly, the effect of NaB treatment consistently showed a significant reduction in viral loads (Fig 5D and 5E). It is important to emphasize that NaB treatment leads to hyperacetylation of all histones. In this respect, we observed hyperacetylation of H3K27, (a CBP target) and H3K9 (a Gcn5 target). However, the acetylation levels of H3K27 were higher than those of H3K9 (S8A and S8B Fig), likely due to the specific enhancement of AaCBP gene expression and activity by ZIKV infection (Fig 2), which was not observed for AaGcn5 activity (Fig 2F, H3K9ac panel). All together, these data suggest a direct correlation between high chromatin decompaction, AMPs overexpression, and strong immunity.

## Discussion

Upon pathogen detection, the innate immune system must be able to mount a robust and quick response, but equally important is the need to rein in the cytotoxic effects of such a response. Immune-response genes are maintained in a silent, yet poised, state that can be readily induced in response to a particular pathogen, and this characteristic pattern is achieved

through the action of two elements: the activation of transcription factors and the modulation of the chromatin environment at gene promoters. Although the activation route of the immune pathways against viruses is relatively well known in *A. aegypti*, our work is the first to attempt to explore the role of chromatin structure in this process.

The regulation of cellular functions by gene activation is accomplished partially by acetylation of histone proteins to open the chromatin conformation, and strikingly, CBP histone acetyltransferase activity always plays a role in this process [39]. One example of signals that ultimately use CBP enzymes as transcriptional coregulators includes the NF-*k*B signaling [40].

*CBP* genes are conserved in a variety of multicellular organisms, from worms to humans and play a central role in coordinating and integrating multiple cell signal-dependent events. In this regard, the *A. aegypti* CBP shows a high degree of homology, notably within the functional domains, with well-characterized human and fly enzymes (Fig 1). Therefore, one might anticipate that AaCBP functions as a transcriptional coactivator in a variety of physiological processes of the mosquito, including innate immunity.

The Toll pathway is an NF-*k*B pathway that is suggested to play a role in immunity in mosquitoes [4,5,34]. Gene expression analysis of ZIKV-infected mosquitoes has shown that Toll pathway-related genes are upregulated in ZIKV infection when compared to other immune pathways [4]. Here, we showed that the expression of AaCBP is also upregulated upon ZIKV infection and that CBP-dependent histone acetylation enables the mosquito to fight viral infections. Although it is not yet clear how AaCBP could limit virus replication, one could envision a molecular role where a particular transcription factor (for example, Rel1) would recruit AaCBP to immune-related gene (for example, AMP genes) promoters and/or enhancers and acetylate histones (for example, H3K27), culminating with chromatin decompaction and gene activation (depicted in our hypothetical model in Fig 6). Indeed, we experimentally demonstrated in part that our model might be correct: 1. We showed that ZIKV-infection potentiates AaCBP-mediated H3K27 acetylation (a mark of gene activation); 2. The lack of AaCBP leads to downregulation of immune-related genes, higher loads of ZIKV virus, and lower rates of mosquito survival; 3. An inverse phenotype was obtained under H3K27 hyperacetylation. Nevertheless, the AaCBP-bound transcription factor in this signaling pathway has yet to be determined.

Host-pathogen interactions provide a highly plastic and dynamic biological system. To cope with the selective constraints imposed by their hosts, many pathogens have evolved unparalleled levels of phenotypic plasticity in their life history traits [41]. Likewise, the host phenotype is drastically and rapidly altered by the presence of a pathogen [42,43]. One important example of these alterations is the manipulative strategy of the pathogen aimed at maximizing its survival and transmission, and one obvious target is the host's immune system. In recent years, the epigenetic modulation of the host's transcriptional program linked to host defense has emerged as a relatively common occurrence of pathogenic viral infections [44]. Interestingly and importantly, our data reveal that chromatin remodeling by histone acetylation contributes to establishing a resistance in ZIKV-infected *A. aegypti*. In this respect, we showed that ZIKV-infected wild type *A. aegypti* resisted to infections better than ZIKV-infected-AaCBP-silenced mosquitoes, whose survival was drastically compromised (Fig 3B). Thus, our data point to an important role of AaCBP in maintaining *A. aegypti* homeostasis through fine-tuning the transcriptional control of immune genes.

Overall, we believe that the disruption of AaCBP would have an effect on immune-gene regulation by compromising its physical interaction with transcription factors and the transcriptional machinery, and/or its enzymatic activity, leading to a significant increase in viral loads. Nevertheless, the precise mechanism that led to the enhancement of ZIKV replication after AaCBP knockdown is still missing. This is particularly true when we compare the kinetics between ZIKV loads (Fig 3C and 3D) and mRNA transcription of AMPs (S6A–S6D Fig.).

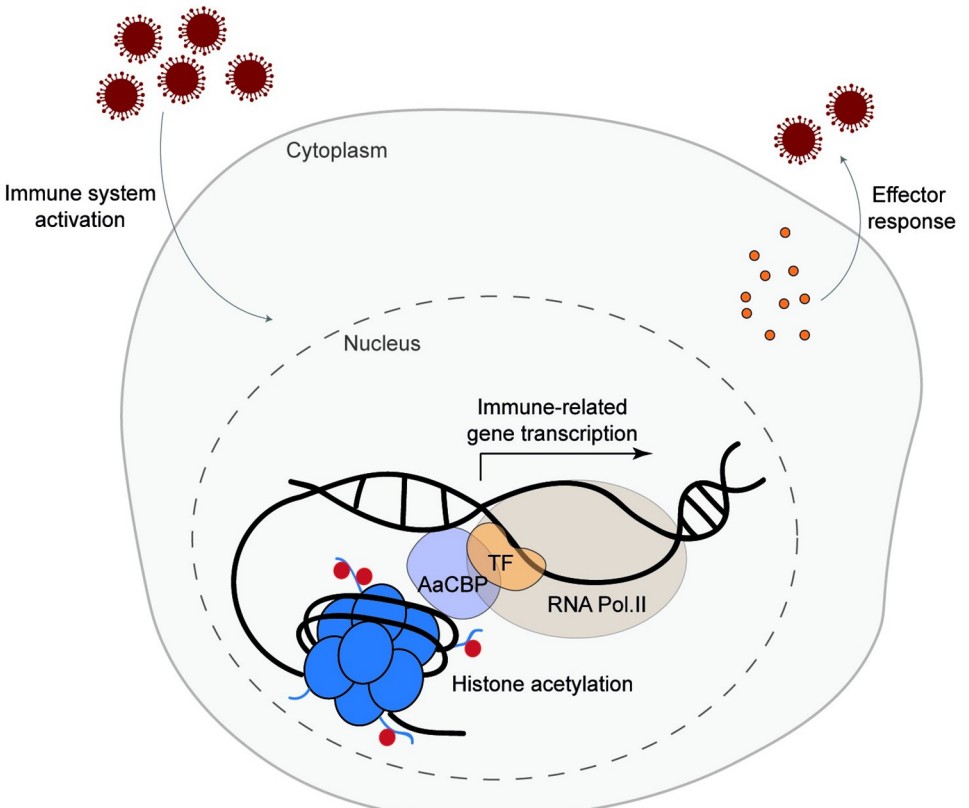

**Fig 6. Proposed model for the role of AaCBP-mediated histone acetylation in suppressing ZIKV infection in *A. aegypti*.** Zika infection signals the functional activation of transcription factors (TF) that translocate to the nucleus where they activate the expression of immune-related genes to fight the virus infection. Chromatin decondensation is mandatory for gene activation. Therefore, AaCBP is likely recruited by TFs to target promoters, where it exerts its histone acetyltransferase activity, leading to an open state of the chromatin at these promoter sites.

Our work has attempted to explore the epigenetic nature of virus-vector interactions. We have focused on epigenetic events that initiate changes in the vector nucleus, likely involving the cooperation between transcription factors and chromatin modifiers to integrate and initiate genomic events, which culminate with the limitation of Zika virus replication.

## Supporting information

**S1 Table. List of primers used in this study.**
(XLSX)

**S1 Fig. Protein domain and alignment of putative histone acetyltransferases from *A. aegypti*.** A. Genome identification of putative lysine acetyltransferase (KAT) homologs from *A. aegypti*, AaHAT1 (XP_001651817.1), AaTip60 (XP_021706851.1), AaMOF1 (XP_001658578.2), AaMOF2 (XP_021711683.1), HBO1 (XP_021701314.1), GCN5 (XP_0016566424.1) and AaCBP itself. Functional domains are indicated above each box. B. Protein sequence alignment of the HAT domains from the putative *A. aegypti* KATs. Amino acids in red show identity or conservation among all 7 HAT domains.
(TIF)

**S2 Fig. Blood meal upregulates the expression of AaCBP.** Mosquitoes were fed with blood over different time courses and mRNA quantification by qPCR was performed in the midgut or fat body. The results in A and B are pools of at least 3 independent experiments, plotted using samples from sugar-fed mosquitoes as reference. Error bars indicate the standard error of the mean; statistical analyses were performed by Student's *t* test. *, $p < 0.05$; **, $p < 0.01$; ***, $p < 0.001$.
(TIF)

**S3 Fig. Silencing levels of AaCBP after a sugar or blood meal.** Two days after feeding, the AaCBP gene was knocked down and its expression was measured two days after silencing. qRT-PCR was performed from 3 independent biological replicates. Bars indicate the standard error of the mean; statistical analyses were performed by Student's *t* test. *, $p < 0.05$.
(TIF)

**S4 Fig. Silencing of AaCBP in adult female mosquitoes by intrathoracic injections of dsRNAs.** The expression (A) and activity (B) of AaCBP were used to evaluate the efficiency of gene knockdown. A. The mRNA levels of AaCBP in the midgut, ovary, fat body or whole mosquito were quantified by qRT-PCR at 48 h postinjection with dsCBP or dsLuc. Silencing level was determined by the ratio between mRNA levels of AaCBP-silenced versus dsLuc-injected mosquitoes. B. Western blot of 10 μg of total protein extract of dsCBP- or dsLuc-injected-mosquitoes. Monoclonal antibodies against acetylated- or nonacetylated histone H3 (loading control) are indicated. The intensity of the bands was quantified by densitometry analysis plotted as a graph using ImageJ (NIH Software). Western blotting was performed on 3 independent biological replicates and one representative is shown here. Bars indicate the standard error of the mean; statistical analyses were performed by Student's *t* test. *, $p < 0.05$.
(TIF)

**S5 Fig. ZIKV infection upregulates the expression of antimicrobial peptides in the midgut of *A. aegypti*.** Fifty adult female mosquitoes were infected with ZIKV after feeding on infected blood and the expression of *cecropin D* (CEC D), *cecropin G* (CEC G), *attacin* (ATTC), *defensin C* (DEF C) and *serpin* (SERP) was measured by qRT-PCR four days post infection. qRT-PCR was performed from 3 independent biological replicates. Bars indicate the standard error of the mean.
(TIF)

**S6 Fig. Silencing of AaCBP is sustainable and efficient until five days of ZIKV infection.** Fifty adult female mosquitoes were infected with ZIKV for 1, 5, 7 or 10 days and the expression levels of AaCBP, *cecropin G*, *defensin C or vago 2* in the fat body were measured by qRT-PCR on 3 independent biological replicates. Bars indicate the standard error of the mean; statistical analyses were performed by Student's *t* test. *, $p < 0.05$.
(TIF)

**S7 Fig. Intestinal microbiota load by genomic DNA qPCR.** Fifty nanograms of genomic DNA from a pool of 15 midguts from silenced female mosquitoes was used as a template. Amplifications were carried out using specific primers for the ribosomal genes 16S and 18S, for bacteria or fungi, respectively. The *A. aegypti* ribosomal protein 49 gene (*Rp*49) was used as an endogenous control. Quantifications were carried out using the comparative *Ct* method. Genomic DNA qPCR was performed from 6 independent biological replicates. Bars indicate the standard error of the mean.
(TIF)

**S8 Fig. Histone deacetylase inhibition leads to histone hyperacetylation in *A. aegypti*.** Ten adult female mosquitoes were intrathoracically injected with PBS, 0.25 M (17.5 pmol), or 0.5

M (35 pmol) of sodium butyrate (NaB). Four days after treatment, 10 µg of total protein extract from 10 mosquitoes was used for histone acetylation analysis. Western blotting with monoclonal antibodies against H3K9ac, H3K27ac, H3 panacetylated, or H3 (as loading control) was performed. The intensity of the bands was quantified by densitometry (lower panel) analysis plotted as a graph using ImageJ (NIH Software). Western blotting was performed on 3 independent biological replicates and one representative is shown in panel A. Error bars in Panel B indicate the standard error of the mean; statistical analyses were performed by Student's *t* test. *, $p < 0.05$.
(TIF)

## Acknowledgments

We thank Jaciara Miranda Freire for running the insectary and excellent technical assistance with mosquito rearing and Dr. Ana Cristina Bahia Nascimento (IBCCF–UFRJ) for productive discussions.

## Author Contributions

**Conceptualization:** Anderson de Mendonça Amarante, Isabel Caetano de Abreu da Silva, Vitor Coutinho Carneiro, Amanda Roberta Revoredo Vicentino, Kanhu Charan Moharana, Pedro Lagerblad de Oliveira, Marcelo Rosado Fantappié.

**Data curation:** Anderson de Mendonça Amarante, Kanhu Charan Moharana, Thiago Motta Venancio, Marcelo Rosado Fantappié.

**Formal analysis:** Marcelo Rosado Fantappié.

**Funding acquisition:** Pedro Lagerblad de Oliveira, Marcelo Rosado Fantappié.

**Investigation:** Anderson de Mendonça Amarante, Isabel Caetano de Abreu da Silva, Vitor Coutinho Carneiro, Amanda Roberta Revoredo Vicentino, Marcelo Rosado Fantappié.

**Methodology:** Anderson de Mendonça Amarante, Isabel Caetano de Abreu da Silva, Marcia de Amorim Pinto, Luiza Mendonça Higa, Octavio A. C. Talyuli.

**Project administration:** Marcelo Rosado Fantappié.

**Resources:** Pedro Lagerblad de Oliveira, Marcelo Rosado Fantappié.

**Software:** Kanhu Charan Moharana, Thiago Motta Venancio.

**Supervision:** Marcelo Rosado Fantappié.

**Validation:** Anderson de Mendonça Amarante, Isabel Caetano de Abreu da Silva, Vitor Coutinho Carneiro, Amanda Roberta Revoredo Vicentino, Octavio A. C. Talyuli, Marcelo Rosado Fantappié.

**Visualization:** Marcelo Rosado Fantappié.

**Writing – original draft:** Anderson de Mendonça Amarante, Marcelo Rosado Fantappié.

**Writing – review & editing:** Anderson de Mendonça Amarante, Marcelo Rosado Fantappié.

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
