## [Decision Letter · Decision Letter 0]

21 Jan 2022

Dear Dr Fantappié,

Thank you very much for submitting your manuscript "Zika virus infection drives epigenetic modulation of immunity by the histone acetyltransferase CBP of Aedes aegypti" for consideration at PLOS Neglected Tropical Diseases. As with all papers reviewed by the journal, your manuscript was reviewed by members of the editorial board and by several independent reviewers. In light of the reviews (below this email), we would like to invite the resubmission of a significantly-revised version that takes into account the reviewers' comments. 

We cannot make any decision about publication until we have seen the revised manuscript and your response to the reviewers' comments. Your revised manuscript is also likely to be sent to reviewers for further evaluation.

Sincerely,

Jason L. Rasgon

Associate Editor

Philippe Desprès

Deputy Editor

Reviewer's Responses to Questions

**Key Review Criteria Required for Acceptance?**

**Methods**

-Are the objectives of the study clearly articulated with a clear testable hypothesis stated?

-Is the study design appropriate to address the stated objectives?

-Is the population clearly described and appropriate for the hypothesis being tested?

-Is the sample size sufficient to ensure adequate power to address the hypothesis being tested?

-Were correct statistical analysis used to support conclusions?

-Are there concerns about ethical or regulatory requirements being met?

Reviewer #1: This study fulfills above mentioned requirements. Please see the summary and general comments section for a detailed review comments.

Reviewer #2: Critical details regarding some methods are missing, including sample sizes, number of replications and statistical tests.

**Results**

-Does the analysis presented match the analysis plan?

-Are the results clearly and completely presented?

-Are the figures (Tables, Images) of sufficient quality for clarity?

Reviewer #1: Yes.

Please see the summary and general comments section for a detailed review comments.

Reviewer #2: Results are clearly presented and figures are good quality

**Conclusions**

-Are the conclusions supported by the data presented?

-Are the limitations of analysis clearly described?

-Do the authors discuss how these data can be helpful to advance our understanding of the topic under study?

-Is public health relevance addressed?

Reviewer #1: Yes. 

Please see the summary and general comments section for a detailed review comments.

Reviewer #2: The conclusions are only partially supported by the data. Some contradictions are not discussed, nor are alternative explanations developed and discussed.

**Editorial and Data Presentation Modifications?**

Reviewer #1: Accept with minor revision

Reviewer #2: (No Response)

**Summary and General Comments**

Reviewer #1: This exciting manuscript investigates the epigenetic modulation of the Zika virus in Ae. aegypti mosquitoes. Authors have performed bioinformatics characterization of CREB-binding protein (CBP) orthologue in Aedes aegypti (AaCBP), and expression pattern of AaCBP in different tissues of ZIKV infected Aedes aegypti mosquitoes. The CBP promotes transcription through chromatin remodeling, acetylation of proteins, and recruitment of the basal transcription machinery. Using pharmacological activator and genetic knockdown strategies, authors have demonstrated a viable model explaining how Zika virus-mediated acetylation of histone proteins moderated chromatin conformation that modulates innate immune response of Aedes aegypti and maintains homeostasis and pathological balance to tolerate persistent infection and develop efficient antiviral strategies to restrict viral replication to non-pathogenic levels. 

Although the experiments are well planned and executed, the following minor but important clarifications/additions would help improve the manuscript:

1. In Figure 4 the authors identified three immune genes that are down-regulated in response to dsRNA KD of AaCBP, namely Cecropin D, Cecropin G, and Defensin C. However, in Figure 5 they did not include Cecropin G for analysis and included Defensin A. An explanation or addition of data for Defensin A in figure 4 would bring more clarity for the readers and consistency to the story.

2. In figure 4 legend, defensin is misspelled. Also, to maintain consistency with the other gene names, please change the first alphabet of gene names ‘dicer’ and ‘argonaute’ into lower case letters. 

3. Figure 6 can be presented in a better way e.g., the cytoplasmic and nuclear compartments need to be separated. The mechanistic details as explained in the results section, figure legend and discussion are missing in the figure.

Reviewer #2: Amarante and colleagues present an analysis of the Ae. aegypti transcription factor and histone acyltransferase, AaCBP, in modulating ZIKV infection. AaCBP was induced by ZIKV infection, and knockdown of AaCBP1 decreased the expression of various immune effectors. Additional data after knockdown showed some increases in virus titer in the mosquito and suggest decreases in overall survival. The authors go on to propose a model whereby ZIKV induces AaCBP, which in turn facilitates the activation of immune genes through modifications in chromatin structure. Overall, the work is highly interesting, but there are some gaps and inconsistencies between what the authors report in their figures and their interpretation. 

Concerns

For survival curves the authors report “much higher mortality” for dsAsCBP-ZIKV group, but no statistical analysis is presented. The legend states that 100 mosquitoes were used per group, but no indication is given as to whether this experiment was repeated (or how many times), the size/type of the cage these mosquitoes were held in, or whether dead individuals were removed on a daily basis. It is not uncommon for mold to grow on the bodies of dead individuals-in a crowded arena this can exacerbate mortality amongst others in the cage which essentially serves as a feedback loop. This can be mitigated by keeping the mosquitoes in many small groups (5 cups of 20 or even better 10 cups of 10, so an artifactual death spiral would only affect a single replicate cup) or by removing dead individuals each day. 

The authors show that silencing of AaCBP is over by 10 DPI, and that levels of AMPs and Vago2 are back to normal by 10 DPI (Fig S6). However, differences in ZIKV titer do not emerge until 10 DPI (Fig 3), with no difference at 5 DPI-even though that is when silencing of AaCBP and the AMPs are maximal. This is not consistent with AaCBP or the indicated AMPs acting directly in anti-ZIKV immunity, and this inconsistency is not addressed in the manuscript even though it goes to the heart of the main conclusions. 

Likewise in Figure 5, NaB experiments show that AMPs are induced in ZIKV-infected individuals at 3 DPI with reduced virus titer at 7 DPI, but not 15 DPI. Again, this is not consistent with the data presented in Fig3, where the authors report an increase in viral titer at 15 DPI after knockdown of CBP. 

In reference to Figure 3, the authors state “We clearly saw that AaCBP-silenced mosquitoes did not efficiently fight virus infections in these tissues that are expected to mount strong immune responses against viruses” – This phrase overstates the findings presented. The authors observed a less than 2-fold increase in virus titer at 10+15 DPI. No difference at all was observed at 5 DPI. This does not mean that these mosquitoes did not mount an immune response, and it is not clear what “efficient” means in this context since that is typically a measure of how much you get out based on how much you put in. Also, the term “clearly” is subjective and should be removed. Given that virus titers in the control group here were about 4 times lower at 15 DPI as compared to the control group in Figure 5 (~2000 PFUs per midgut vs 8000 PFUs per midgut), the authors need to be much more cautious in their interpretations since there is clearly substantial variation between experiments. This variation is never explained at since the authors are injecting virus should not be present (that is the whole point of injections, to make dose uniform).

The authors state that “An increase in total histone H3 acetylation could not be observed (Supplementary Fig 7A and B), which is somehow expected if one considers that a combination of acetylated and nonacetylated H3 might coexist in a specific cell type and/or during a specific period.” – This does not make sense. Acetylation can occur at H3 positions K9, K14, K18, K23 and K27. The authors document what appears to be an increase in acetylation at positions K9 and K27. Total acetylation did not appear to change. Provided the authors data are reliable, the most obvious possibility to check is that acetylation at some of the other positions has decreased. If cell-type specific events were obfuscating total acetylation, they should obscure K9 or K27-specific acetylation as well. Of course, as the authors state, NaB should increase acetylation across all sites, so this is still a bit inconsistent with the authors conclusions. 

“The Toll pathway has been shown to play the most important role in controlling ZIKV infections (Angleró-Rodríguez et al., 2017).” -This is not an accurate representation of the findings of Anglero-Rodriguez. In that paper, the authors found the strongest anti-viral response when the Toll pathway was artificially stimulated as compared only with IMD and Jak/Stat. These experiments did not consider any other immune pathway, of which there are still many.

The authors do not adequately consider other hypotheses that could better explain their findings, making it difficult to come to the same conclusions as are reached in the model presented in Figure 6. Two points that need to be further developed (either experimentally or in the discussion or both) are potential affects on the microbiome and the more general role of CBP outside of its role as a HAT. For the former, dysregulation of AMPs may lead to dysbiosis or greater susceptibility to fungal pathogens-maybe this can be compensated for alone but not with the added stress of ZIKV infection. In the latter, dsRNA experiments cannot disentangle the role of CBP in histone acetylation from its more general role as CREB binding protein in RNA transcription.

PLOS authors have the option to publish the peer review history of their article (what does this mean?). If published, this will include your full peer review and any attached files.

Reviewer #1: Yes: Sujit Pujhari

Reviewer #2: No
---

## [Decision Letter · Decision Letter 1]

31 May 2022

Dear Dr FANTAPPIE,

Thank you very much for submitting your manuscript "Zika virus infection drives epigenetic modulation of immunity by the histone acetyltransferase CBP of Aedes aegypti" for consideration at PLOS Neglected Tropical Diseases. As with all papers reviewed by the journal, your manuscript was reviewed by members of the editorial board and by several independent reviewers. The reviewers appreciated the attention to an important topic. Based on the reviews, we are likely to accept this manuscript for publication, providing that you modify the manuscript according to the review recommendations. 

Thank you for your revision. Reviewer 2 still has some issues that need to be addressed (and I believe they can be done quickly). After you address them, I do not anticipate needing to send this out for further review and will process the resubmission with all due speed.

Sincerely,

Jason L. Rasgon

Associate Editor

Philippe Desprès

Deputy Editor

Thank you for your revision. Reviewer 2 still has some issues that need to be addressed (I believe they can be done quickly). After you address them, I do not anticipate needing to send this out for further review.

Reviewer's Responses to Questions

**Key Review Criteria Required for Acceptance?**

**Methods**

-Are the objectives of the study clearly articulated with a clear testable hypothesis stated?

-Is the study design appropriate to address the stated objectives?

-Is the population clearly described and appropriate for the hypothesis being tested?

-Is the sample size sufficient to ensure adequate power to address the hypothesis being tested?

-Were correct statistical analysis used to support conclusions?

-Are there concerns about ethical or regulatory requirements being met?

Reviewer #1: Yes

Reviewer #2: Some key information on methods are still missing.

**Results**

-Does the analysis presented match the analysis plan?

-Are the results clearly and completely presented?

-Are the figures (Tables, Images) of sufficient quality for clarity?

Reviewer #1: Yes

Reviewer #2: (No Response)

**Conclusions**

-Are the conclusions supported by the data presented?

-Are the limitations of analysis clearly described?

-Do the authors discuss how these data can be helpful to advance our understanding of the topic under study?

-Is public health relevance addressed?

Reviewer #1: Yes

Reviewer #2: (No Response)

**Editorial and Data Presentation Modifications?**

Reviewer #1: (No Response)

Reviewer #2: (No Response)

**Summary and General Comments**

Reviewer #1: In this revised manuscript Amarante et al. have satisfactorily addressed the concerns pointed out by the reviewers. Overall, this work is exciting and analyses the role of epigenetic modulations of histone acetyltransferase CBP in Zika virus interaction in Ae. ageypti. The authors propose a model where ZIKV infection activates TF and induces AaCBP resulting in chromatin decondensation and activation of immune genes.

Reviewer #2: Reviewer #2:

The legend states that 100 mosquitoes were used per group, but no indication is given as to

whether this experiment was repeated (or how many times), the size/type of the cage these

mosquitoes were held in,

Authors:

All experiments in this study were repeated at least three times. The survival curves

were repeated 5 times. This information is provided in the legend of Figure 3B (page

28).

New Comment: No information on replicates is present in the legend of Fig3B. 

or whether dead individuals were removed on a daily basis. It is not uncommon for mold to

grow on the bodies of dead individuals-in a crowded arena this can exacerbate mortality

amongst others in the cage which essentially serves as a feedback loop. This can be

mitigated by keeping the mosquitoes in many small groups (5 cups of 20 or even better 10

cups of 10, so an artifactual death spiral would only affect a single replicate cup) or by

removing dead individuals each day.

Authors:

Yes, we agree, and we were aware of this potential problem. The survival

experiments were conducted in five 500ml cups, each containing 20 mosquitoes. Dead

mosquitoes were removed on a daily basis.

New Comment: This is re-assuring, but this information is not included in the revised manuscript.

Reviewer #2:

The authors show that silencing of AaCBP is over by 10 DPI, and that levels of AMPs and

Vago2 are back to normal by 10 DPI (Fig S6). However, differences in ZIKV titer do not

emerge until 10 DPI (Fig 3), with no difference at 5 DPI-even though that is when silencing

of AaCBP and the AMPs are maximal. This is not consistent with AaCBP or the indicated

AMPs acting directly in anti-ZIKV immunity, and this inconsistency is not addressed in the

manuscript even though it goes to the heart of the main conclusions. Likewise in Figure 5,

NaB experiments show that AMPs are induced in ZIKV-infected individuals at 3 DPI with

reduced virus titer at 7 DPI, but not 15 DPI. Again, this is not consistent with the data

presented in Fig3, where the authors report an increase in viral titer at 15 DPI after

knockdown of CBP.

Authors:

The data of Figure S6 show the levels of mRNA transcripts, which are indeed

compromised until Day 5 and start to be restored at Day 10. However, it is difficult to

tell when proteins are fully expressed, translated and/or active to re-establish cellular

homeostasis. The same consideration can be assumed for the data of Figure 5.

In addition, please, see the data of Figure 2D and E. Note that in Aag2 cells, AaCBP

gene transcripts reach their peak at 6 hours post-infection, but the peak of its

enzymatic activity is only reached at 15 hours post-infection.

New Comment: A delay between RNA levels and protein levels may be able to explain a difference of hours, but not days. AMPs by their nature must be produced rapidly and then vanish when the threat is gone. No matter what the explanation, the manuscript must reflect these inconsistencies, rather than ignore them. 

Reviewer #2:

In the latter, dsRNA experiments cannot disentangle the role of CBP in histone acetylation

from its more general role as CREB binding protein in RNA transcription.

Authors:

We are not aware of the function of CBP proteins (in all different model organisms thus far studied) outside its role as a transcription coactivator through its histone acetyltransferase activity. However, this is a clear possibility, and this possibility can certainly be addressed in our future studies.

New Comment: I am confused by this response, as the authors themselves describe CBPs as being “recruited to promoters by interaction with DNA-bound transcription factors, which directly interact with the RNA polymerase II complex.”. Thus, CBP is a physical scaffold protein, and the authors also state “The TAZ, KIX and CREB are the protein interaction domains that mediate interactions with transcription factors”. Thus, silencing of CBP depleted not just the HAT function, but also the scaffold function. This should be discussed. 

New comment:

“These results indicate that even a partial deletion of AaCBP is enough to disrupt the homeostasis of the mosquito”- The phrase “partial deletion” here does not make sense, as nothing was deleted. Replace with “partial reduction”, “partial silencing” or similar phrase. Maybe the authors meant “partial depletion”?

PLOS authors have the option to publish the peer review history of their article (what does this mean?). If published, this will include your full peer review and any attached files.

Reviewer #1: Yes: Sujit Pujhari

Reviewer #2: No

Figure Files:

Data Requirements:

Reproducibility:

References

---

## [Editor Report · Decision Letter 2]

3 Jun 2022

Dear Dr FANTAPPIE,

We are pleased to inform you that your manuscript 'Zika virus infection drives epigenetic modulation of immunity by the histone acetyltransferase CBP of Aedes aegypti' has been provisionally accepted for publication in PLOS Neglected Tropical Diseases.

Best regards,

Jason L. Rasgon

Associate Editor

Philippe Desprès

Deputy Editor

---

## [Editor Report · Acceptance letter]

21 Jun 2022

Dear Dr. Prof. Fantappié,

We are delighted to inform you that your manuscript, "Zika virus infection drives epigenetic modulation of immunity by the histone acetyltransferase CBP of Aedes aegypti," has been formally accepted for publication in PLOS Neglected Tropical Diseases.

Best regards,

Shaden Kamhawi

co-Editor-in-Chief

Paul Brindley

co-Editor-in-Chief
